# Influence of Fiber Content on Shear Capacity of Steel Fiber-Reinforced Concrete Beams

**Juan Andres Torres** [1] and **Eva O.L. Lantsoght** [1,2,*]

1   Politécnico, Universidad San Francisco de Quito, Quito EC 170157, Ecuador; jtorresj@estud.usfq.edu.ec
2   Engineering Structures, Civil Engineering and Geosciences, Delft University of Technology, 2628 CN Delft, The Netherlands
*   Correspondence: e.o.l.lantsoght@tudelft.nl

**Abstract:** For shear-critical structural elements where the use of stirrups is not desirable, such as slabs or beams with reinforcement congestion, steel fibers can be used as shear reinforcement. The contribution of the steel fibers to the shear capacity lies in the action of the steel fibers bridging the shear crack, which increases the shear capacity and prevents a brittle failure mode. This study evaluates the effect of the amount of fibers in a concrete mix on the shear capacity of steel fiber-reinforced concrete beams with mild steel tension reinforcement and without stirrups. For this purpose, 10 beams were tested. Five different fiber volume fractions were studied: 0.0%, 0.3%, 0.6%, 0.9%, and 1.2%. For each different steel fiber concrete mix, the concrete compressive strength was determined on cylinders and the tensile strength was determined in a flexural test on beam specimens. Additionally, the influence of fibers on the shear capacity was analyzed based on results reported in the literature, as well as based on the expressions derived for estimating the shear capacity of steel fiber-reinforced concrete beams. The outcome of these experiments is that a fiber percentage of 1.2% or fiber factor of 0.96 can be used to replace minimum stirrups according to ACI 318-14 and a 0.6% fiber volume fraction or fiber factor of 0.48 to replace minimum stirrups according to Eurocode 2. A fiber percentage of 1.2% or fiber factor of 0.96 was observed to change the failure mode from shear failure to flexural failure. The results of this study support the inclusion of provisions for steel fiber-reinforced concrete in building codes and provides recommendations for inclusion in ACI 318-14 and Eurocode 2, so that a wider adoption of steel fiber reinforced concrete can be achieved in the construction industry.

**Keywords:** experiments; fiber factor; fiber volume fraction; flexure; shear; steel fiber reinforced concrete

## 1. Introduction

The behavior of steel fiber-reinforced concrete (SFRC) structural elements under flexure and shear has been a topic of research for the last decades [1]. It is important to understand the influence of fibers on the shear capacity in structural elements such as beams or slabs, because providing fibers can be an efficient solution in elements where there is reinforcement congestion or where the geometry complicates the use of some or all stirrups. Several studies mention that the addition of steel fibers enhances the mechanical properties of the concrete such as its tensile strength [2,3], ductility [2], and toughness [2]. Moreover, it has been observed that adding steel fibers to reinforced concrete can lead to a failure mode change from a brittle shear failure to a flexural failure when the load is applied at the shear-critical position [4].

At the present time, code provisions are limited regarding the sectional strength of SFRC. For instance, ACI 318-14 [5] does not include provisions for the shear capacity of SFRC, but a minimum fiber content of 0.75% is permitted when the conditions provided in Equation (1) are satisfied.

$$0.5\phi V_c < V_u < \phi V_c$$
$$\text{when } V_u < \phi 0.17 \sqrt{f_c'} b_w d \text{ with } f_c' \text{ in MPa} \tag{1}$$
$$\text{and } h < 600 \text{ mm}$$

As a result, structural applications of SFRC in countries where the North American codes are governing are limited. On the other hand, a number of national codes with shear provisions for SFRC exist, for example the French code [6], German guidelines [7], and Italian code [8]. The *fib* model code 2010 [9] also includes the use of SFRC and contains provisions for the shear capacity of SFRC.

Experimental research [10] findings lead to the conclusion that, depending on the type of fiber and fiber material properties, a good performance in terms of shear capacity, ductility and crack control can be achieved with a fiber content of 1.0%. However, a fiber content of 0.75% is recommended to be used as replacement of the minimum stirrup reinforcement based on experimental observations on normal-strength concrete [11] and according to the requirements of the ACI Committee 318-14 [5]. Furthermore, a fiber content greater than 1.0% does not seem to improve significantly the shear capacity of SFRC [11]. The effect of the fiber content on the failure mode has been studied as well. A series of experiments [12,13] led to the conclusions that a fiber content of 0.5% is required to change the failure mode from shear to flexure for the studied material properties and boundary conditions.

The aim of this study is to extend knowledge on the shear capacity of SFRC in order to allow a wider use of SFRC in structural elements. In particular, our study focuses on the influence of the amount of fibers used in the mix, and its relation to the shear capacity. The study is limited to the shear capacity of normal strength (20 to 35 MPa) concrete. For this purpose, we tested 10 SFRC beams with mild steel longitudinal reinforcement and without stirrups. The testing program included specimens with fiber percentages from 0.0% to 1.2%. The beams were subjected to four-point bending. The sectional shear force at inclined cracking and at the ultimate are analyzed to determine the contribution of fibers to the shear capacity. The outcome of the experiments also served for comparison of the experimental results with the currently existing expressions for the shear capacity of SFRC. Additionally, we compared our experimental observations with trends observed in a database of shear experiments on SFRC beams from the literature [14].

## 2. Existing Models for the Shear and Flexural Capacity of Steel Fiber-Reinforced Concrete (SFRC)

In this section, we summarize the currently available models for determining the shear and flexural capacity of SFRC. We used these expressions to prepare the experiments presented in this article, for the interpretation and analysis of the experimental results, and to derive recommendations for the use of a certain fiber volume fraction.

### 2.1. Ultimate Shear Capacity

There are several theories that describe the shear behavior of reinforced concrete such as the Modified Compression Field Theory (MCFT) [15] based on equilibrium conditions, compatibility requirements, and stress-strain relationships, or the Critical Shear Crack Displacement Theory (CSDT) proposed by Yang [16], which takes into account the different shear-carrying mechanisms after cracking (aggregate interlock, dowel action, and concrete in the compression zone). The original version of the CSDT does not take into account the contribution of fibers. However, Filian et al. [17] extended the CSDT to take into account the capacity of steel fibers to carry tension across the crack as an additional shear-resisting mechanism.

Different expressions to determine the shear capacity of SFRC beams are provided in Table 1. Research conducted by Lee et al. [18] extends the concept of the Dual Potential Capacity Model

(DPCM), proposed in previous studies [19–21] by the authors, to SFRC. To calculate the shear demand in reinforced concrete, the DPCM considers aggregate interlock in the cracked tension zone and in the compression zone, as well as crack widths in the tension zone. However, when applied to SFRC, the model only considers the capacity in the tension zone based on the crack width. The contribution of fibers is taken into account based on the Direct Tension Force Transfer Model (DTFTM), proposed by the authors in their previous research [22–28], the random distribution of the steel fibers, and the pull-out strength of the steel fibers. The ultimate shear strength of the SFRC beam is calculated by summing the minimum shear contribution of the concrete (i.e., intersection between demand and capacity at compression and tension zone) and the contribution of fiber as determined in Equation (3).

To calculate the shear capacity of SFRC, mostly (semi)-empirical expressions are used. Most expressions take into account the properties and geometry of the fibers, often in the form of the fiber factor *F*, a concept proposed by Narayanan and Palanjian [29]. The fiber factor *F* is calculated as follows:

$$F = \frac{L}{D} V_f D_f \tag{2}$$

where $L$ = length of the fiber. $D$ = diameter of the fiber. $V_f$ = fiber volume fraction. $D_f$ = fiber bond factor.

The fiber bond factor ($D_f$) accounts for the geometry and bond characteristics of the fibers. For steel fibers, it has a value of 1.00 for hooked fibers, 0.75 for crimped fibers and 0.50 for straight fibers as recommended by Narayanan and Darwish [30]; another method to calculate the fiber bond factor is suggested by [31,32] and it is equal to the ratio between the mean fiber–matrix shear stress and the strength in direct tension of the material. This approach is useful when fibers of materials different than steel are used.

Different parameters are considered in the equations summarized in Table 1, for instance the aggregate size factor, which considers the maximum aggregate size, is considered in Equations (7), (12) and (13). Imam et al. [33] studied the effect of adding fibers in simply reinforced high-strength concrete beams without stirrups and its influence on flexure/shear interaction. The authors proposed an equation to predict the ultimate strength of SFRC based on the simultaneous occurrence of arching action and shear-resisting mechanisms, considering the equilibrium of forces in the shear span at the ultimate state. The expression is a function of the fiber factor ($F$) and the longitudinal steel ratio ($\rho$) considered in one term ($\omega$), the effect of relative beam size to the maximum aggregate size ($d/d_a$), and the aggregate size ($d_a$), which is considered in the size effect term ($\psi$). Yakoub [34] provides two different equations to predict the shear capacity of slender SFRC beams ($a/d > 2.5$). The first equation (Equation (12)) is a modification to include the effect of steel fibers of the shear capacity proposed by Bažant and Kim [35] for normal-strength reinforced concrete. The expression takes into account the size of aggregates ($d_a$), the concrete compressive strength ($f_c$), shear span to depth ratio ($a/d$), and longitudinal reinforcement ratio ($\rho$). The second equation by Yakoub [34], Equation (13) is an extension of the expression for the shear capacity of the Canadian Code CSA A23.3-04 [36] to include the contribution of the steel fibers. This expression is a function of the strain at mid-depth of the beam ($\varepsilon_x$) and crack spacing ($s_x$) as a function of the aggregate size ($d_a$). Equation (13) does not consider arching action.

Combining the concrete contribution and the fiber contribution to find the shear resistance is an approach followed by a number of authors. Dinh et al. [37] conducted an experimental program which resulted in an expression to estimate the shear strength provided by the fibers in SFRC beams without stirrups based on the tensile strength of plain fiber-reinforced concrete prisms and the measured crack widths according to the standard ASTM C-1609 [38]. The model proposed by Dinh et al. [37] combines the contribution of the fibers, evaluated as the vertical component of the tensile strength from the fibers bridging the crack, which depends on the crack width, with the concrete contribution, determined as the shear contribution of the concrete in the compression zone. The fiber contribution is a function of the crack width ($w$). An equivalent uniform tensile stress ($f_f$) is used to find the force resultant of

the fiber contribution. The result of these procedures is that the ultimate shear strength is calculated by the summation of Equations (9) and (10). Similarly, Mansur et al. [39] conducted an experimental program to provide an expression to predict the shear capacity of SFRC by adding the contribution of fibers ($V_{sf}$) to the concrete contribution ($V_c$) as calculated in Equation (15). Both Dinh et al. [37] and Mansur et al. [39] use similar expressions for $V_{sf}$ and include similar parameters such as the tensile strength of concrete ($f_t$), the geometry of the beam, and the diagonal crack angle (taken as 30 degrees by [37] and 45 degrees by [39]). On the other hand, the expressions for the concrete contribution are based on different assumptions. Dinh et al. [37] consider a uniform shear stress over the depth of the compression zone, whereas Mansur et al. [39] consider the ratio of external shear to moment according to the recommendation of the ACI-ASCE Committee 426 [40].

Empirical equations have been developed and validated through experimental programs. Narayanan and Darwish [30] developed Equation (17) for the ultimate shear strength by testing SFRC beams with different crimped fiber contents and fiber aspect ratios of 100 and 133, with variable *a/d* ratio and concrete compressive strengths from 36 to 75 MPa. A similar experimental program [13] with two different compressive strengths (31 and 65 MPa) and hooked-end steel fibers with an aspect ratio of 62.5 was used to develop Equation (19). Moreover, Shin et al. [41] developed Equation (20) by testing 22 reinforced concrete beams with and without steel fibers and with a concrete compressive strength of 80 MPa. The main variables in this program were the fiber content, *a/d* ratio, amount of longitudinal reinforcement, and amount of shear reinforcement. All of the proposed equations consider three shear-resisting mechanisms: (1) the fiber contribution represented by the splitting cylinder strength $f_{sp}$, (2) dowel action provided by the longitudinal reinforcement and taking into account the influence of the shear span to depth ratio, and (3) the fiber pullout stresses along the inclined cracks, $v_b$. Arching action is taken into account by using the factor *e*, but small differences exist between Equations (17) and (19), and the effect of arching action is not considered in Equation (20).

A second set of empirical expressions takes into account the concrete compressive strength ($f_c$), fiber factor ($F$), longitudinal reinforcement ratio ($\rho$), and shear span to depth ratio (*a/d*). Based on testing high strength ($f_c$ about 93 MPa) SFRC beams with variable hooked-end steel fiber (aspect ratio of 75) content, longitudinal reinforcement ratio, and shear span to depth ratio (*a/d*), Ashour et al. [42] developed two expressions: (1) Equation (21), an extension of Zsuty's equation [43] to include the contribution of the fibers through the fiber factor *F*, and (2) Equation (22), an extension of the ACI 318-89 [44] shear equation to include the contribution of the fibers, as well as the effect of the shear span to depth ratio and the longitudinal reinforcement ratio. The factor 0.7 accounts for the action of high strength concrete. Khuntia et al. [45] developed Equation (23) based on 10 different experimental programs in which the main variables were concrete compressive strength ($f_c$), shear span to depth ratio (*a/d*), fiber factor (*F*), fiber content ($V_f$), and longitudinal reinforcement ratio ($\rho$). The expression sums the concrete contribution from ACI 318-95 [46] and the contribution of the fibers, assuming a diagonal crack of 45 degrees. The arching action that is developed when *a/d* is less than 2.5 is taken into account in the factor $\alpha$.

A different approach is followed by Kara [47], who used gene expression programming (GEP) to predict the ultimate shear strength of SFRC beams without stirrups. A database of 101 tests was used to build the GEP model with five main variables: concrete compressive strength ($f_c$), effective depth (*d*), shear span to depth ratio (*a/d*), longitudinal reinforcement ratio ($\rho$), and fiber factor (*F*). The model resulted in Equation (24) were the coefficients $c_0$, $c_1$, $c_2$, and $c_3$ are constants provided by the formulation of the GEP model.

**Table 1.** Expressions for predicting the ultimate shear capacity of steel fiber-reinforced concrete (SFRC) beams without stirrups.

| Authors | Ref | Expression | Equation |
|---|---|---|---|
| Lee et al. | [18] | $V_{sf} = 0.41 F \tau_{\max} b_w (d - c) \cot \theta$ <br> with $\tau_{\max} = 0.85 \sqrt{f_c}$ <br> $V_c = \min(\frac{0.18\lambda\sqrt{f_c}}{0.31 + 0.686 w_s} b_w(d-c), 0.52\sqrt{f_c} b_w c)$ <br> $V_u = V_{sf} + V_c$ | (3) <br> (4) <br> (5) <br> (6) |
| Imam et al. | [33] | $V_u = 0.6\psi \sqrt[3]{\omega}\left[ f_c^{0.44} + 275 \sqrt{\frac{\omega}{(a/d)^5}} \right] b_w d$ <br> with $\omega = \rho(1 + 4F)$ <br> $\psi = \frac{1 + \sqrt{5.08/d_a}}{\sqrt{1 + d/(25 d_a)}}$ | (7) |
| Arslan | [48] | $V_u = \left( 0.2 f_c^{2/3}\left(\frac{c}{d}\right) + \sqrt{\rho(1 + 4F) f_c} \right)\left(\frac{3.0}{a/d}\right)^{1/3} b_w d$ | (8) |
| Dinh et al. | [37] | $V_c = 0.11 f_c \beta_1 c b_w = 0.13 A_s f_y$ <br> with $\beta_1$ from Whitney's stress block <br> $V_{sf} = (\sigma_t)_{avg} b_w(d-c)\cot(\theta)$ <br> $(\sigma_t)_{avg}$ is the average tensile stress of SFRC <br> $V_u = V_{sf} + V_c$ | (9) <br><br> (10) <br><br> (11) |
| Yakoub | [34] | $V_u =$ <br> $\left[ 0.83\xi\sqrt[3]{\rho}\left( \sqrt{f_c} + 249.28\sqrt{\frac{\rho}{(a/d)^5}} \right) + 0.162 F\sqrt{f_c} \right] b_w d$ <br> with $\xi = 1/\sqrt{1 + d/(25 d_a)}$ <br> $V_u = \beta\sqrt{f_c}(1 + 0.70F) b_w d$ <br> with $\beta = \frac{0.4}{1 + 1500\varepsilon_x}\frac{1300}{1000 + s_{xe}}$ <br> $\varepsilon_x = \frac{M/d_v + V}{2 E_s A_s}$ <br> $d_v = \max(0.9d, 0.72h)$ <br> $s_{xe} = \frac{35 s_x}{16 + d_a} \geq 0.85 s_x$ | (12) <br><br><br><br> (13) |
| Mansur et al. | [39] | $V_{sf} = \sigma_t b_w d$ <br> with $\sigma_t = 0.68\sqrt{f_c}$ <br> $V_c = \left( 0.16\sqrt{f_c} + 17.2\frac{\rho V d}{M} \right) b_w d > (0.29\sqrt{f_c}) b_w d$ <br> $V_u = V_{sf} + V_c$ | (14) <br><br> (15) <br> (16) |
| Narayanan and Darwish | [30] | $V_u = \left[ e\left( 0.24 f_{sp} + 80\rho\frac{d}{a} \right) + v_b \right] b_w d$ <br> $e = 1$ when $\frac{a}{d} > 2.8$ <br> $e = 2.8\frac{d}{a}$ when $\frac{a}{d} \leq 2.8$ <br> with $f_{sp} = \frac{f_{cuf}}{20 - \sqrt{F}} + 0.7 + 1.0\sqrt{F}$ | (17) <br><br><br> (18) |
| Kwak et al. | [13] | $V_u = \left[ 3.7 e f_{sp}^{2/3}\left( \rho\frac{d}{a} \right)^{1/3} + 0.8 v_b \right] b_w d$ <br> $e = 1$ when $\frac{a}{d} > 3.4$ <br> $e = 3.4\frac{d}{a}$ when $\frac{a}{d} \leq 3.4$ <br> with $f_{sp}$ from Equation (18) | (19) |
| Shin et al. | [41] | $V_u = \left( 0.22 f_{sp} + 217\rho\frac{d}{a} + 0.834 v_b \right) b_w d$ <br> with $f_{sp}$ from Equation (18) | (20) |
| Ashour et al. | [42] | $V_u = \left[ (2.11\sqrt[3]{f_c} + 7F)\left( \rho\frac{d}{a} \right)^{0.333} \right] b_w d$ for $\frac{a}{d} \geq 2.5$ <br> $V_u = \left[ (0.7\sqrt{f_c} + 7F)\frac{d}{a} + 17.2\rho\frac{d}{a} \right] b_w d$ | (21) <br> (22) |
| Khuntia et al. | [45] | $V_u = \left[ (0.167\alpha + 0.25F)\sqrt{f_c} \right] b_w d$ <br> with $\alpha = 2.5\frac{d}{a} < 3$ for $a/d < 2.5$ <br> $\alpha = 1$ for $a/d > 2.5$ | (23) |
| Kara | [47] | $V_u = \left[ \left( \frac{\rho d}{c_0 c_1 (a/d)} \right)^3 + \frac{F d^{1/4}}{c_2} + \sqrt{\frac{c_3 f_c}{d}} \right] b_w d$ <br> with $c_0 = 3.324; c_1 = 0.909; c_2 = 2.289; c_3 = 9.436$ | (24) |

### 2.2. Sectional Shear at Inclined Cracking Load

Table 2 gives the expressions to determine the sectional shear at inclined cracking. Arslan [48] initially provided Equation (8) to capture the contribution of the fibers to the ultimate shear strength of SFRC slender beams, considering the increase of stiffness in the dowel zone due to the presence of the fibers. Later research [49] provided an equation for the inclined cracking load by introducing a strength reduction factor of 0.6, as shown in Equation (25). Narayanan and Darwish [30] provided Equation (26) based on their experimental observations, following the same format as their expression for the ultimate shear capacity, Equation (17), except that arching action is not accounted for. A simpler equation, Equation (27), is provided by Kwak et al. [13]. This expression does not consider the fiber factor *F*. It only considers the splitting cylinder strength and the dowel action provided by the longitudinal reinforcement $\rho$ and the *a/d* ratio.

**Table 2.** Expressions for predicting the inclined cracking load in SFRC beams without stirrups.

| Authors | Ref | Expression | Equation |
|---|---|---|---|
| Arslan et al. | [49] | $V_{cr} =$ $0.6\left(0.2f_c^{2/3}\left(\frac{c}{d}\right) + \sqrt{\rho(1+4F)f_c}\right)\left(\frac{3.0}{a/d}\right)^{1/3}b_w d$ | (25) |
| Narayanan and Darwish | [30] | $V_{cr} = \left(0.24f_{sp} + 20\rho\frac{d}{a} + 0.5F\right)b_w d$ | (26) |
| Kwak et al. | [13] | $V_{cr} = \left(3f_{sp}^{2/3}\sqrt[3]{\rho\frac{d}{a}}\right)b_w d$ | (27) |

### 2.3. Flexural Capacity

The flexural capacity is calculated based on horizontal equilibrium, taking into account the contribution of the fibers. Compatibility of strains is assumed, and stress-strain relationships are introduced to find the stresses and resulting forces. Imam et al. [33] proposed an expression based on the horizontal equilibrium, with assumptions for the tensile and compressive stress blocks as shown in Figure 1. The nominal flexural moment is then calculated according to Equation (28). The same equilibrium and strain compatibility assumptions as in Figure 1 but a different shape for the tensile stress block were used to develop Equation (29), which determines the nominal flexural moment capacity for SFRC beams [39]. A newer expression for determination of the flexural capacity is introduced by [50] in which the residual strength of SFRC and depth of the tensile zone is accounted as expressed in Equation (30). An additional reference on flexural capacity can be consulted in [51].

$$M_n = \frac{1}{2}\rho f_y b_w d^2 (2 - \eta) + 0.83 F b_w d^2 (0.75 - \eta)(2.75 + \eta) \tag{28}$$

$$\text{with } \eta = \frac{\rho f_y + 2.32F}{0.85 f_c + 3.08F}$$

$$M_n = A_s f_y(d - c) + \sigma_t b_w \frac{(h - c)^2}{2} + 0.85 f_c a b_w \left(c - \frac{a}{2}\right) \tag{29}$$

$$\text{with } \sigma_t = 0.68\sqrt{f_c}$$

$$M_u = \left[\rho f_y\left(1 - 0.5\frac{0.80c}{d}\right) + f_r\left(\frac{h - e}{d}\right)\left(\frac{h}{d} - \frac{h - e}{2d} - 0.5\frac{0.80c}{d}\right)\right]b_w d^2 \tag{30}$$

$$\text{with } e = c\frac{\frac{f_{ctf}}{E_{ct}} + \varepsilon_{o85}}{\varepsilon_{o85}}$$

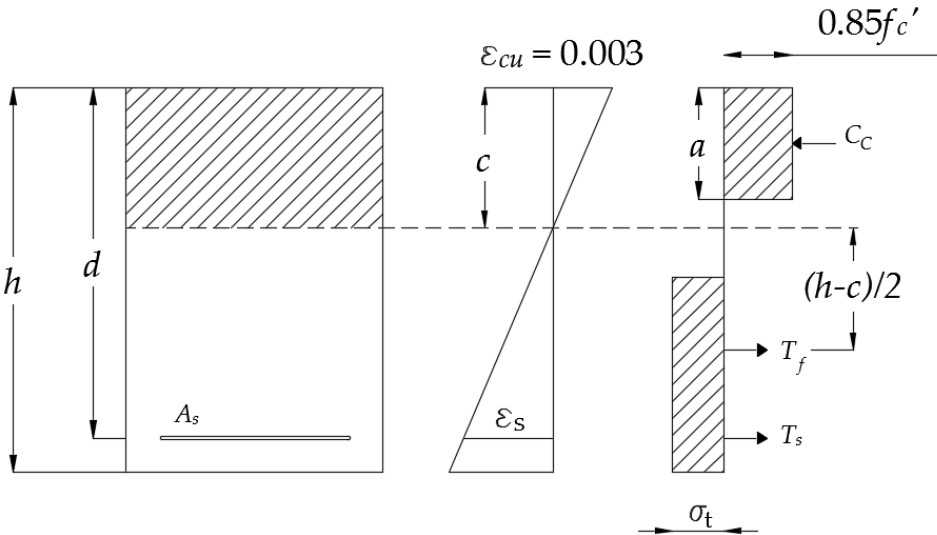

**Figure 1.** Equilibrium and assumption of forces for flexural analysis.

## 3. Materials and Methods

### 3.1. Materials

Concrete mix design was done based on ACI 211.1 [52] for the reference mix without fibers. This reference mix was then adjusted accordingly for the increasing fiber contents. Table 3 provides the concrete mix proportions for all mixes used in this study (with different fiber volume fractions). The fiber percentages correspond to a volume fraction of all the materials. The mix design was carried out to obtain normal strength concrete.

**Table 3.** Mix design.

| Fiber Content (%) | Cement (kg/m$^3$) | Fine Aggregates (kg/m$^3$) | Coarse Aggregates (kg/m$^3$) | Water (kg/m$^3$) | Steel Fibers (kg/m$^3$) | w/cm | Fiber Factor |
|---|---|---|---|---|---|---|---|
| 0.0 | 575 | 875 | 585 | 253 | - | 0.40 | 0.00 |
| 0.3 | 557 | 848 | 567 | 273 | 23.6 | 0.45 | 0.24 |
| 0.6 | 555 | 845 | 565 | 272 | 47.1 | 0.45 | 0.48 |
| 0.9 | 538 | 820 | 548 | 291 | 68.7 | 0.50 | 0.72 |
| 1.2 | 508 | 792 | 518 | 319 | 94.4 | 0.55 | 0.96 |

The same constituent materials were used in all the mixes. The cement used was Type IP, which is a blended portland-pozzolan cement that meets the requirements of ASTM C 595 [53]. The coarse aggregates are crushed andesite igneous stone. The maximum aggregate size is 9.5 mm. For the fine aggregates, material passing the No. 4 sieve is used (i.e., sand). No additives were used in any mix. The steel fibers used in all the mixes are hooked-end fibers with an aspect ratio of 80. These fibers were provided by Bekaert and the commercial name of the fiber type is Dramix 3D [54]. The properties of the steel fibers used in the experiments are given in Table 4 and a picture of the steel fibers is shown in Figure 2.

**Table 4.** Steel fiber properties [55].

| Property | Value |
|---|---|
| Length | 60 mm |
| Diameter | 0.75 mm |
| Tensile strength | 1225 MPa |
| Modulus of Elasticity | 210,000 MPa |
| Shape | hooked-end |

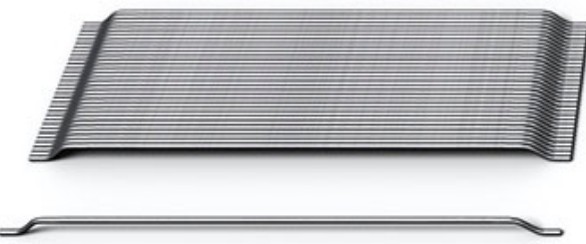

**Figure 2.** Dramix 3D steel fibers [55].

We used trial batches of the mixes to find the optimal workability. The mix design from Table 3 is the final mix design, which was used for casting the beams. It can be seen that the higher the fiber content is, the higher the required water to cementitious material ratio (w/cm) is, because high fiber contents affect the workability of the concrete. All specimens were compacted on a vibration table.

Longitudinal reinforcement of 16 mm diameter was used in all reinforced beams. The steel grade is 42 according to the Ecuadorian INEN standard 2167 [56], which means that the characteristic yield strength is 420 MPa. To determine the properties, tensile tests were carried out on three samples of the reinforcing steel by an external laboratory. The results of these tests are provided in Table 5 and an estimated simplified stress–strain diagram is shown in Figure 3.

**Table 5.** Reinforcement steel properties.

| Property | Value |
|---|---|
| Nominal diameter | 16 mm |
| Yield Strength | 452 MPa |
| Ultimate Strength | 601 MPa |
| Modulus of Elasticity | 176,667 MPa |

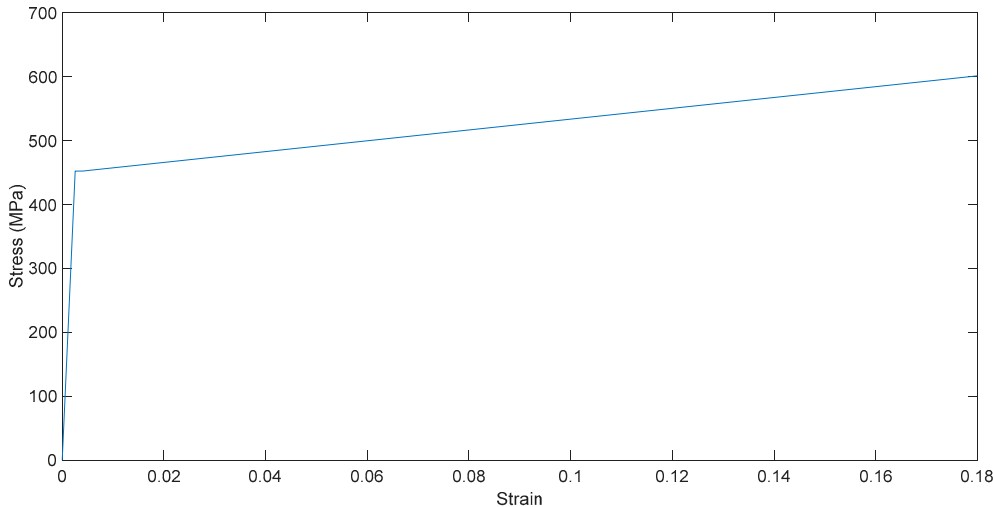

**Figure 3.** Estimated stress–strain diagram of the reinforcing steel.

The material properties are determined through compression tests on concrete cylinders and flexural tests on prisms. For each fiber content, three 200 mm × 100 mm cylinders are cast for compression tests according to standard ASTM C39 [57], and two prisms of 200 mm × 200 mm × 600 mm are cast for testing according to standard ASTM C1609 [38], see Figure 4. The resulting material properties from the concrete compressive and tensile strength tests are shown in Table 6 and a selection of load-displacement diagrams is shown in Figure 5. The difference in the slope of the first branch is due to slip of the LVDT (linear variable differential transformer) at the beginning of the loading process. For the 0.3% fiber content, one of the tensile stress tests resulted in a very low peak flexural stress even lower than the average 0.0%, thus when using a fiber content of 0.3% the distribution of the fibers highly affects the mechanical properties of the material. In all specimens with fibers we observed a stiffening behavior after development of the first crack. To quantify this effect, we divided the peak load stress by the first peak stress (i.e., flexural stress at first crack). This effect is indicated as "Tension stiffening capacity" in Table 6. For the beams with a 1.2% fiber content the maximum capacity of the testing machine was reached prior to failure; the maximum load is reported instead. As a result, the tension-stiffening capacity of the 1.2% fiber content mix cannot be calculated.

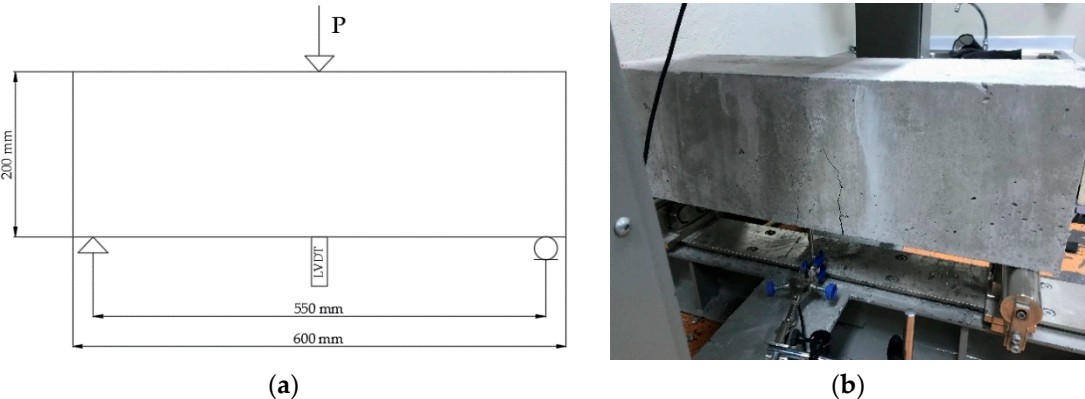

**Figure 4.** (**a**) Sketch of the setup for tensile strength test. (**b**) Failure of specimen in tensile strength test.

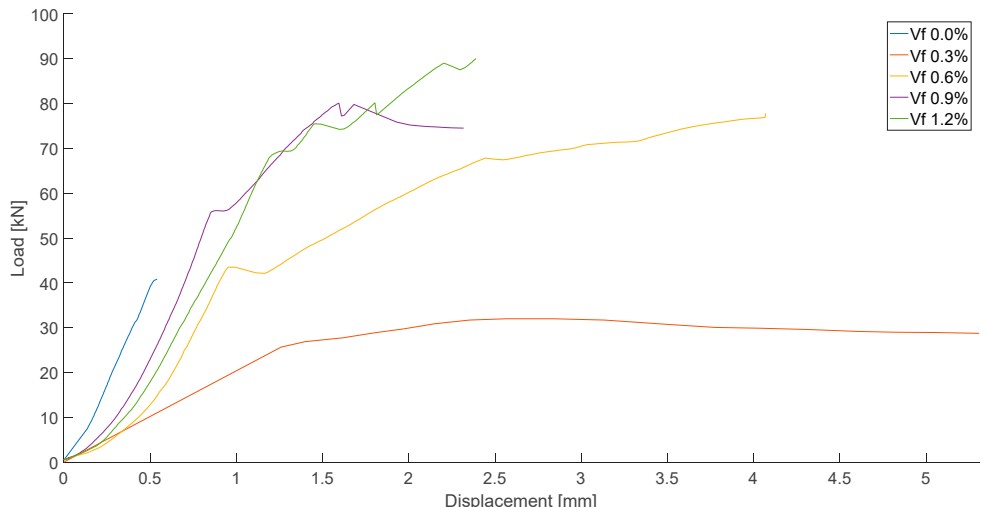

**Figure 5.** Load-displacement diagrams of tensile strength tests.

**Table 6.** Hardened concrete properties.

| Fiber Content (%) | Compressive Strength (MPa) | Flexural Stress at First Peak (MPa) | Deflection at First Peak (mm) | Peak Flexural Stress (MPa) | Peak Deflection (mm) | Tension Stiffening Capacity |
|---|---|---|---|---|---|---|
| 0.0 | 20.6 | - | - | 2.88 | 0.600 | - |
| 0.3 | 33.0 | 1.77 * | 1.260 * | 2.82 | 1.820 | 1.25 * |
| 0.6 | 27.8 | 2.86 | 0.637 | 5.39 | 3.676 | 1.88 |
| 0.9 | 29.1 | 3.38 | 0.857 | 6.00 | 2.103 | 1.78 |
| 1.2 | 30.3 | 5.35 | 1.024 | 6.16 | 1.942 | - |

* Values obtained from test on one prism only.

### 3.2. Test Setup and Instrumentation

The beam specimens are designed to achieve a shear failure prior to a flexural failure. As such, they are over-reinforced for flexure. The design procedure was an iterative process evaluating all the equations previously stated in Table 1 and taking the maximum shear capacity and the minimum flexural capacity given by Equations (28) and (29) for flexure, and Equations (3)–(16) for shear. The remaining equations were added later to verify their accuracy in the predictions. For the design of the experiment, we estimated the concrete compressive strength as 28 MPa, which was the target value for the mix design. Table 7 provides the design flexural and shear capacities with the respective equations, and the associated load for the calculated sectional shear and sectional moment capacity. The associated load for achieve a flexural failure remains the same for all the fiber contents because it is more dependent on the longitudinal reinforcement and the fibers do not have a large effect on the flexural capacity. The resulting reinforcement ratio of the longitudinal reinforcement is $\rho = 4.02\%$, which allows us to study the mechanism of failure of shear. An anchorage system was used because the development length of the bars did not fit within the beam; the system consisted of anchorage steel plates with a thickness of 15 mm that were welded to the bars.

**Table 7.** Design shear and flexure capacities, and associated loads.

| Fiber Content (%) | Maximum $V_u$ [Equation] (kN) | Associated Load (kN) | Minimum $M_n$ [Equation] (kN-m) | Associated Load (kN) |
|---|---|---|---|---|
| 0.0 | 21.3 [(12)] | 42.6 | 10.9 [(28) and (29)] | 76.4 |
| 0.3 | 24.5 [(8)] | 49.0 | 10.9 [(28)] | 76.4 |
| 0.6 | 27.8 [(8)] | 55.6 | 10.9 [(28)] | 76.4 |
| 0.9 | 30.6 [(8)] | 61.2 | 10.9 [(28)] | 76.4 |
| 1.2 | 33.1 [(8)] | 66.2 | 10.9 [(28)] | 76.4 |

Figure 6 shows a sketch, the cross section and a photograph of the test setup. A four-point bending test was carried out. The resulting shear span to depth ratio (*a/d*) was 2.85, for which a shear failure is expected. The loading plate size is 260 mm × 150 mm. The beam is supported by rollers with a length of 300 mm and a diameter of 40 mm. The width of the contact surface can be estimated as 10 mm. The load is applied in a displacement-controlled manner with a speed of 0.006 mm/s until failure. For each experiment, two LVDTs are used to measure displacements: one under the load and the second one between the load and the support (in the shear span). Additionally, a camera is used for future analysis of the photographs with digital image correlation (DIC).

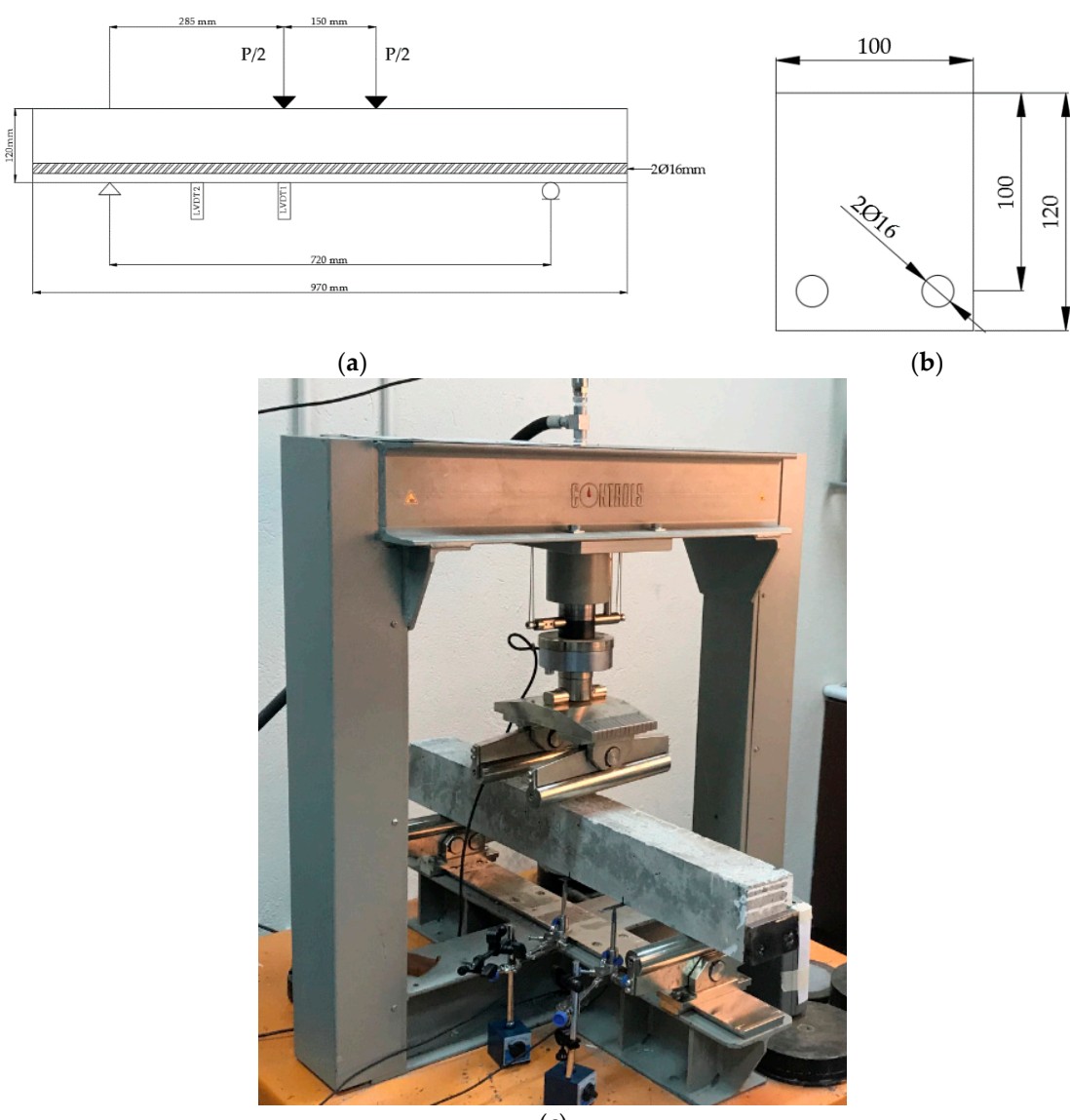

**Figure 6.** (**a**) Sketch of the setup of the experiment. (**b**) Cross-section of the beam for shear experiments (all units in mm) [58]. (**c**) Picture of the setup.

## 4. Results

### 4.1. Experimental Results

Ten reinforced beams are tested in four-point bending as sketched in Figure 6. Figure 7 shows a selection of load-displacement diagrams for the tested specimens. The reported displacement in these diagrams is measured by the LVDT placed under the load. A first peak can be seen when inclined cracking occurs for all the specimens. It is important to mention that we also observed this first peak for the specimens not containing steel fibers. This observation may indicate that arching action was developed and that the failure mode of the beams was a shear-compression failure.

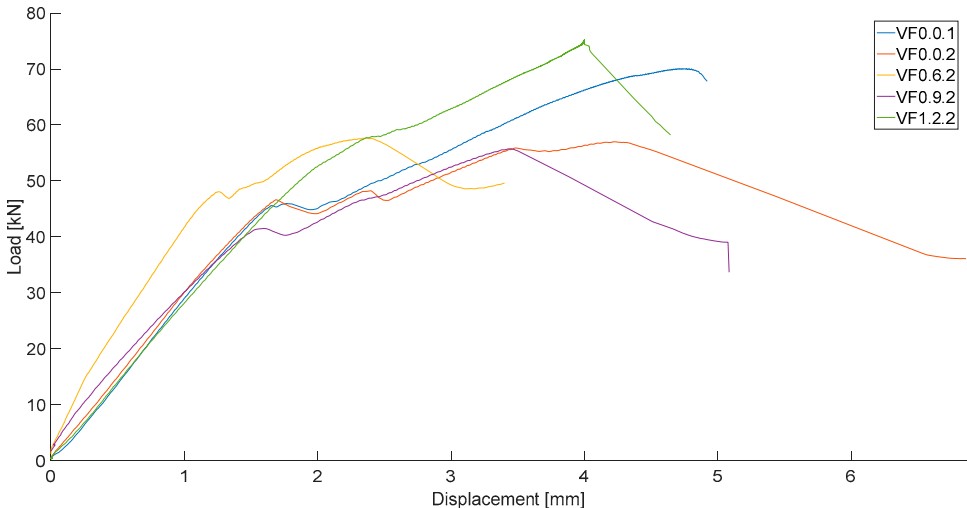

**Figure 7.** Load-displacement diagrams for a selection of the tests.

Table 8 shows the experimental results of the inclined cracking load ($P_{cr}$), the load that was applied at the moment of failure ($P_u$), the maximum sectional shear force calculated by the sum of the sectional shear caused by the applied load and the self-weight of the beam (which can be considered negligible) ($V_u$), the normalized shear stress, the deflection at failure ($\delta_u$), and the failure mode that occurred for each test.

**Table 8.** Results and failure mode for 10 SFRC beam tests.

| Specimen ID | Fiber Content (%) | $P_{cr}$ (kN) | $P_u$ (kN) | $V_u$ (kN) | $\dfrac{V_u}{(b_w d \sqrt{f_c})}$ | $\delta_u$ (mm) | Failure Mode |
|---|---|---|---|---|---|---|---|
| VF0.0.1 | 0.0 | 45.74 | 70.1 | 35.20 | 0.772 | 4.738 | Shear |
| VF0.0.2 | 0.0 | 46.77 | 57.0 | 28.65 | 0.628 | 4.235 | Shear |
| VF0.3.1 | 0.3 | 47.78 | 61.7 | 31.00 | 0.537 | 3.030 | Shear |
| VF0.3.2 | 0.3 | 46.99 | 66.8 | 33.55 | 0.581 | 1.603 * | Shear |
| VF0.6.1 | 0.6 | 54.62 | 68.1 | 34.20 | 0.646 | 2.606 † | Shear |
| VF0.6.2 | 0.6 | 48.20 | 57.7 | 29.00 | 0.547 | 2.372 | Shear |
| VF0.9.1 | 0.9 | 48.48 | 62.5 | 31.40 | 0.579 | 4.000 | Shear |
| VF0.9.2 | 0.9 | 41.64 | 55.8 | 28.05 | 0.517 | 3.445 | Shear |
| VF1.2.1 | 1.2 | 56.50 | 68.1 | 34.20 | 0.619 | 1.919 ‡ | Shear |
| VF1.2.2 | 1.2 | 57.88 | 75.2 | 37.75 | 0.683 | 4.000 | Shear + Flexure |

* Deflection at inclined cracking load. † Deflection at failure in the shear span. ‡ Deflection at failure in the shear span.

For a fiber percentage of 1.2%, which is associated with a fiber factor of 0.96, we observe a change from a shear failure to a shear-flexural failure. This effect is seen in specimen VF1.2.2: during testing, the flexural cracks are visible and opening, and at failure, some signs of crushing of the concrete are seen as well. The observed cracks are not visible in the other tests, for which the failure load is equal to or larger than the failure load in VF1.2.2 (i.e., VF0.0.1 and VF0.6.1) as shown in Figure 8.

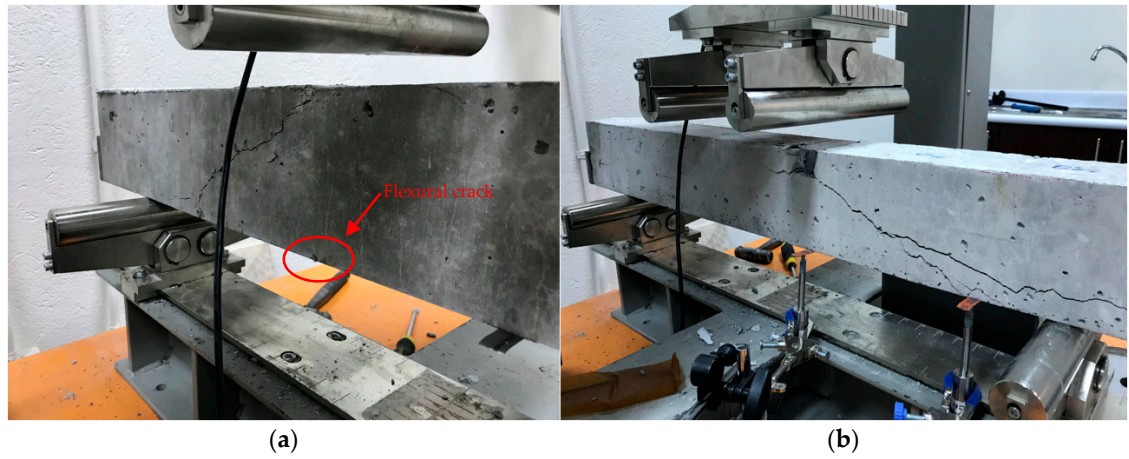

**Figure 8.** (**a**) VF1.2.2 after failure and (**b**) VF0.0.1 after failure.

*4.2. Comparison to Predicted Shear Capacities*

An analysis of the accuracy of the different methods of prediction for ultimate shear capacities and inclined cracking capacities is presented in Tables 9 and 10, respectively. The results are presented in terms of the average tested/predicted shear capacities and their associated standard deviation and coefficient of variation. All the equations underestimated the ultimate shear capacities and inclined cracking capacities, except Equation (20) which overestimated the ultimate shear capacity. Based on these indicators, the method that most closely predicts the ultimate shear capacity is Equation (19) by Kwak et al. [13] with an average tested/predicted value of 1.209 an associated standard deviation of 0.421 and coefficient of variation of 34.8%. The expression that shows the least variability on the ratio of the tested to predicted result is Equation (20) by Shin et al. [41], which gave an average tested to predicted shear capacity of 0.744, with a standard deviation of 0.113 and a coefficient of variation of 15.2%: However, this expression considerably overestimated the ultimate shear capacity. The expression by Yakoub [34], Equation (12), performed well: the average tested to predicted shear capacity was 1.289, with a standard deviation of 0.214 and coefficient of variation of 16.6%.

**Table 9.** Comparison between experimental results and prediction of ultimate shear capacities of SFRC beams.

| Authors | Equation | Average Tested/Predicted | Standard Deviation | Coefficient of Variation |
|---|---|---|---|---|
| Lee et al. | (6) | 1.864 | 0.499 | 0.268 |
| Imam et al. | (7) | 1.839 | 0.577 | 0.314 |
| Arslan | (8) | 1.244 | 0.373 | 0.230 |
| Dinh et al. | (11) | 1.701 | 0.932 | 0.548 |
| Yakoub | (12) | 1.289 | 0.214 | 0.166 |
|  | (13) | 1.764 | 0.343 | 0.195 |
| Mansur et al. | (16) | 1.978 | 0.795 | 0.402 |
| Narayanan and Darwish | (17) | 1.301 | 0.432 | 0.332 |
| Kwak et al. | (19) | 1.209 | 0.421 | 0.348 |
| Shin et al. | (20) | 0.744 | 0.113 | 0.152 |
| Ashour et al. | (21) | 1.476 | 0.493 | 0.334 |
|  | (22) | 1.351 | 0.603 | 0.446 |
| Khuntia et al. | (23) | 2.394 | 1.081 | 0.452 |
| Kara | (24) | 1.432 | 0.420 | 0.294 |

**Table 10.** Comparison between experimental results and prediction of inclined cracking capacities of SFRC beams.

| Authors | Equation | Average Tested/Predicted | Standard Deviation | Coefficient of Variation |
|---------|----------|--------------------------|--------------------|--------------------------|
| Arslan | (25) | 1.579 | 0.417 | 0.264 |
| Narayanan and Darwish | (26) | 2.096 | 0.598 | 0.255 |
| Kwak et al. | (27) | 1.661 | 0.255 | 0.154 |

For the inclined cracking shear all the equations underpredicted the capacity, even though those expressions consider different factors. The relative small size of the specimens may be the reason why a higher inclined cracking capacity was achieved. The results show that Equation (25) provided the closest results with an average tested/predicted value of 1.579 and a standard deviation of 0.417 and coefficient of variation of 26.4%. Nevertheless, Equation (27) is the one with the least variability with an standard deviation of 0.255 and coefficient of variation of 15.4%. For Equation (27) the average tested/predicted value is 1.661, which is close to the value of Equation (25). As a result we can conclude that Equation (27) has a better performance. No recommendations can be given regarding the prediction of a change in the failure mode from shear to flexure, but it was seen that with a fiber content of 1.2% a transition between these types of failure happened.

### 4.3. Analysis of Influence of Fiber Content on Shear Capacity

Experiments reported in the literature allow us to analyze the influence of adding steel fibers to reinforced concrete on the shear capacity of beam elements. A review [59] of the influence of the fiber content of SFRC mentions that the effect of the fibers is dependent on a variety of factors, such as the fiber aspect ratio, mechanical anchorage, and fiber tensile strength, and that minimum shear reinforcement can be replaced by SFRC with hooked-end steel fibers with a fiber content of 0.75%. This addition leads to an increase of the shear strength above $0.3\sqrt{f_c'}$ as stated on Section 1. While this recommendation is formulated in terms of a fiber volume fraction, it may be preferable to derive a recommendation based on the fiber factor, since *F* considers the different sizes, shapes, and aspect ratios of the different types of steel fibers.

The results of the experiments are shown from Figures 9–11. Figure 9 shows the relation between the inclined cracking load and the fiber content, as well as the curves of predictions. An increase of the inclined cracking shear is seen as the fiber content increases, except for the specimens with a fiber fraction of 0.9%, which resulted in the lowest inclined cracking load of all experiments. An increase of 24% in the inclined cracking capacity is seen for increasing the fiber content from 0.0% to 1.2%. Figure 10a,b show the relation between the normalized ultimate shear stress, the fiber content and the fiber factor, respectively. As seen in these figures, the highest ultimate normalized shear stress corresponds to the specimens with no added fibers. As explained previously, arching action was developed in all the specimens and the addition of fibers does not affect the ultimate shear strength when direct load transfer is presented. The relation between the added shear capacity (i.e., the difference between the normalized ultimate shear stress and the normalized cracking shear stress) and the fiber factor is shown in Figure 11. From these results no relation is observed between these two parameters, since the results do not follow a trend.

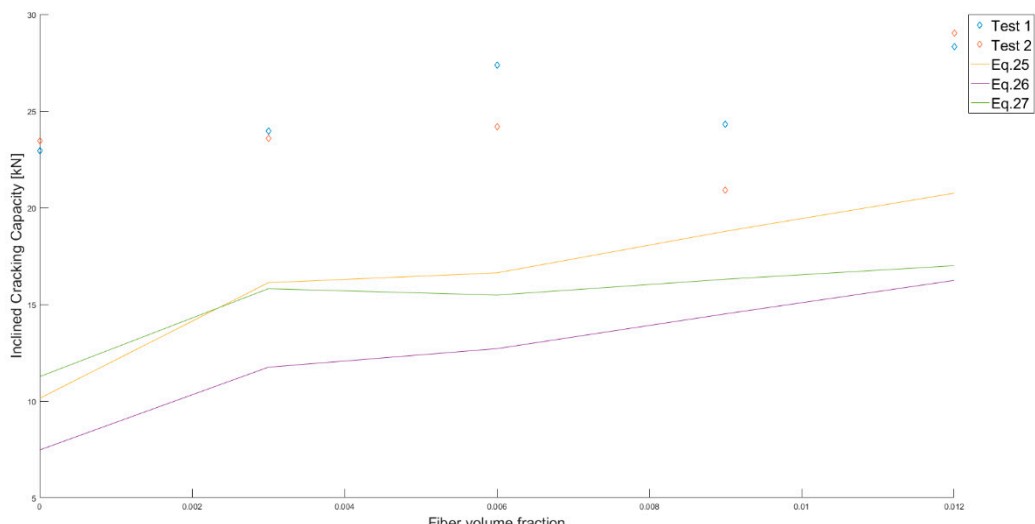

**Figure 9.** Inclined cracking capacity vs. fiber volume fraction, measurements and predictions.

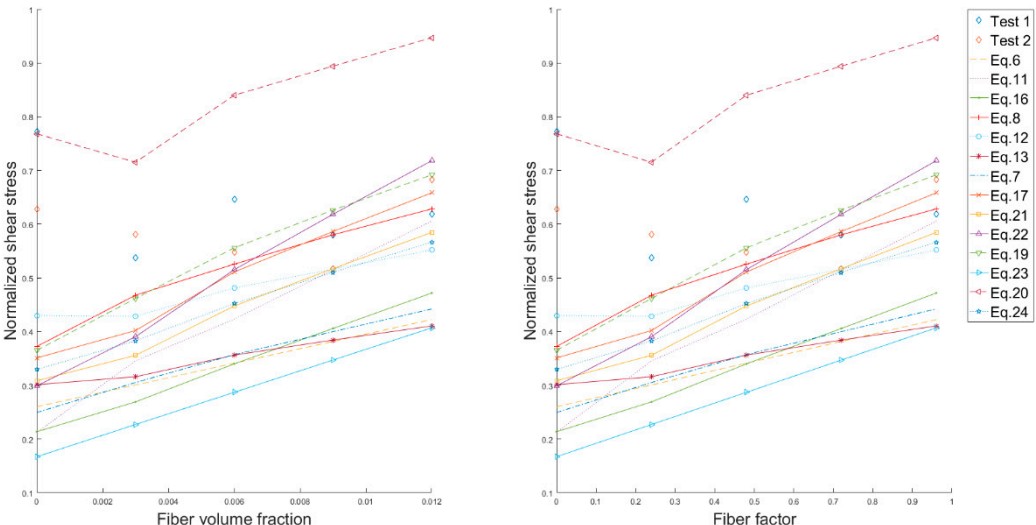

**Figure 10.** Normalized shear stress for ultimate shear capacity vs. fiber volume fraction and vs. fiber factor, predictions and measurements.

A database [14] available in the literature collected the information of 488 experiments of SFRC beams failing in shear. Trends presented in this database show that the normalized shear stress increases with the fiber volume fraction or the fiber factor. This observation differs from what we observed in our experiments, since it seems that fibers do not have an influence on the ultimate shear stress when arching action is developed. When analyzing only the specimens with small *a/d* ratio (less than 3.0) from the database, a similar trend is observed: the higher the fiber factor or fiber content, the higher the normalized ultimate shear stress. Nonetheless, the regression analysis results in a $R^2$ value of 0.1363, which show that the influence of fibers is not very representative as shown in Figure 12. The presence of fibers has an influence on this parameter which is different from the results of our experimental program. However, our experiments follow this trend when the inclined cracking load is considered. These observations further underline the need for a better understanding of the mechanics of the different shear-carrying contributions in SFRC, so that recommendations for fiber contents can be based on sound mechanical concepts.

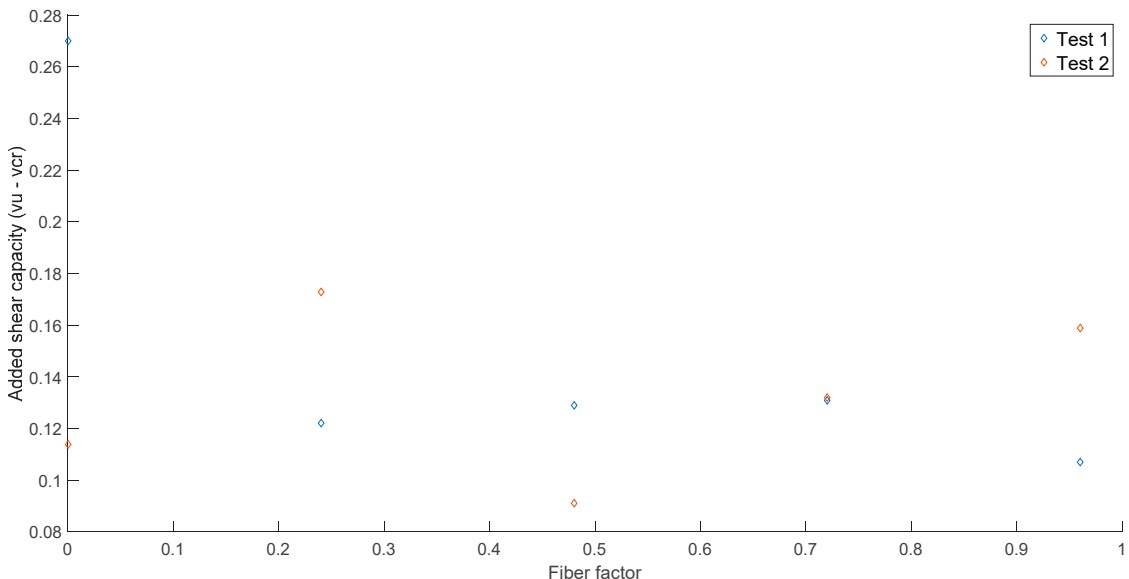

**Figure 11.** Added shear capacity based on normalized shear stress vs. fiber factor.

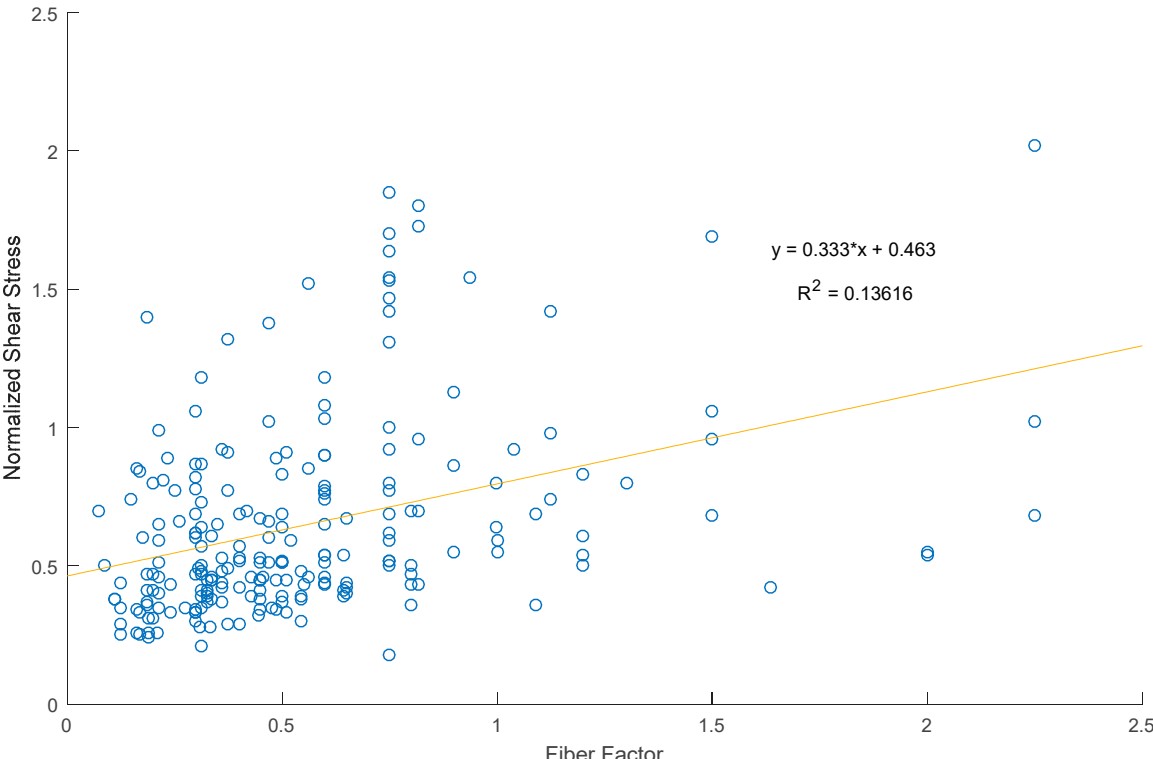

**Figure 12.** Normalized shear stress vs. fiber factor, database trend for small *a*/*d* values.

It is important to mention that the analysis carried out with the database experiments [14] resulted in the same equation (i.e., Equation (19) provided by Kwak et al. [13]) as the one that results in the best predictions and even with a lower average tested/predicted than the value calculated from our experiments.

## 5. Discussion

The results obtained from the different tests provide important information regarding the shear capacity in SFRC beams without stirrups. As previously stated [2–4], the incorporation of fibers enhances the mechanical properties of reinforced concrete. In our material testing, we observed—as expected—a higher tensile strength as the fiber content increased, except for a fiber content of 0.3%, which gave a lower tensile strength than for the specimen with a fiber volume fraction of 0.0%. Tension stiffening was observed in all specimens with steel fibers. Additionally, for higher fiber contents we observed lower peak deflections.

Observations from the shear tests show that, effectively, when steels fibers are provided to reinforced concrete, the failure mode changes from a brittle shear failure to a ductile flexural failure. This observation is important for structural elements where stirrups are not desired. By incorporating steel fibers such shear brittle failure can be prevented and reinforcement congestion can be reduced.

An important observation is that arching action developed in all the beams. This observation explains why the beams with 0.0% fiber content achieved an ultimate capacity similar to those with 1.2%, and why the normalized shear stress of the 0.0% beams is higher than the 1.2%. As such, the addition of steel fibers does not have a significant influence on the ultimate shear strength when arching action develops, i.e., for beams with a short shear span to depth ratio that have an enhanced shear capacity. A review [59] mentions that this effect is increased when steel fibers are added to reinforced concrete.

On the other hand, our results show that there is an influence of the steel fibers on the inclined cracking load as expected; the higher the fiber content, the higher the inclined cracking load, except for the 0.9% fiber content beams. Additionally, from the results obtained from the experiments we can conclude that there is no large influence of the concrete compressive strength on the ultimate shear capacity of SFRC because one of the specimens with the lowest concrete compressive strength (i.e., specimen VF0.0.1) achieved one of the highest values for the ultimate shear capacity. In other words, a beam with a low concrete compressive strength did not result in a low ultimate shear capacity for the range of concrete compressive strengths in our study. The reader should note that the goal of our experimental work was not to study the influence of the concrete compressive strength and we only tested specimens with a target compressive strength of 28 MPa.

As shear resistance consists of a series of shear-resisting mechanisms, research is needed to study the influence of steel fibers on the different mechanisms such as aggregate interlock and dowel action. The contribution of dowel action as a shear resisting mechanism is expected to be relatively larger than it is in reinforced concrete because the presence of randomly distributed steel fibers increases the tensile strength and delays spalling of the concrete cover, upon which dowel action is lost. Moreover, as all the specimens were over-reinforced dowel action is expected to be a mechanism that can significantly carry shear. However, the shear resisting mechanisms such as aggregate interlock and dowel action in SFRC need further research to understand their contribution to the overall shear resistance of a section. The prediction methods considered in this research mostly underestimated the ultimate shear capacity as well as the inclined cracking capacity of the section. Only one method overestimated the ultimate shear capacity. The formulation that best predicted the ultimate shear capacity was Equation (19) proposed by Kwak et al. [13], which resulted in an average tested/predicted shear capacity of 1.209 with an associated standard deviation of 0.421 and coefficient of variation of 34.8%. The approach by Kwak et al. [13] accounts for the arching action by incorporating the factor $a/d$. Other expressions (Equations (8) and (12)) that closely predict the experimental shear capacity also consider this effect. Indeed, Equation (12) had a good performance with an average tested/predicted shear capacity of 1.289, with an associated standard deviation of 0.214 and coefficient of variation of 16.6%. However, the equation that provided the least variability, Equation (20), with a coefficient of variation of 15.2% is the only one that overestimated the ultimate shear capacity. Moreover, Equation (25) provided the best average tested/predicted inclined shear capacity with a value of 1.579. A better overall performance was provided by Equation (27), which resulted in a coefficient of variation of 15.4% on the tested to

predicted shear capacity. Moreover, most of the expressions have better predictions when considering a fiber content of 0.6% or higher. Combining this observation with the low tensile strength measured on the specimens with a 0.3% volume fraction of fibers indicates that the contribution of the fibers in a mix with only 0.3% fibers is not reliable as it is highly dependent on the local distribution of the fibers.

Regarding practical application of SFRC mixes, our experimental results show that a fiber volume fraction of 1.2% or a 0.96 fiber factor can be used to replace the additional shear capacity provided by minimum stirrups according to ACI 318-14 [5]. As compared to the requirements for minimum stirrups from Eurocode 2 [60], all the fiber percentages can be used as a replacement for minimum stirrup. However, a 0.6% fiber content or fiber factor of 0.48 is recommended for the reason that a lower fiber content is highly dependent on the local distribution of fibers.

Our research focuses on the use of steel fibers as a replacement for minimum transverse reinforcement according to different building codes. In seismic design, the transverse reinforcement is usually higher than the minimum and it is used also for confinement of the concrete, so the use of steel fibers alone would not be advisable in this case. However, as steel fibers provide ductility in the case of a shear failure, it would be recommendable to use an optimized mix of stirrups and steel fibers in places where there is reinforcement congestion such as joints. Moreover, research conducted [61–64] in order to study the influence of steel fibers in SFRC subjected to cyclic loading shows that the inclusion of fibers can improve in a satisfactory manner the cyclic response of SFRC members by providing an efficient ductile behavior, higher energy-dissipation capacities, and lower damage indexes compared to reinforced concrete.

## 6. Conclusions

Incorporation of steel fibers in concrete has been observed to enhance the mechanical properties of concrete such as the tensile strength [2–4]. Since shear strength is related to the diagonal tension, adding steel fibers to reinforced concrete members has been observed to increase the shear capacity and sometimes change the failure mode to a ductile flexural failure. Semi-empirical expressions to determine the ultimate shear capacity and inclined cracking capacity have been provided in the literature. These expressions consider different parameters and aim to have a wider use of SFRC in structural elements. Nowadays, different codes [6–9] have provisions for shear in SFRC. Nonetheless, other codes such as ACI 318-14 [5] do not have such provisions.

An experimental program was conducted on 10 SFRC beams with fiber contents of 0.0%, 0.3%, 0.6%, 0.9% and 1.2% to study the contribution of steel fibers to the shear capacity of SFRC, and to compare the experimental results to different proposed equations for the shear capacity of SFRC reported in the literature. Complementary concrete compressive and tensile strength tests were conducted for each fiber content to determine the material properties.

The findings of the experimental results provided information about the shear behavior of SFRC. For a 1.2% fiber content (fiber factor of 0.96), we observed a change of failure mode from shear to shear-flexure, with visible flexural cracks opening prior to failure and indications of local concrete crushing at failure. Moreover, arching action developed in all the shear tests, and from the results we can imply that arching action is not affected by the addition of steel fibers, since one of the specimens without fibers reached a higher failure load than one of the specimens with the highest fiber content in the experimental program (i.e., 1.2%). Nevertheless, when the inclined cracking load is considered, the effect of adding steel fibers is important and results showed that when using a higher fiber content, a higher inclined cracking load is reached with an increase of 24% in shear capacity for a fiber volume fraction increasing from 0.0% to 1.2%.

From the analysis of the different prediction methods, we found that most of the expressions considered tend to underestimate the ultimate shear capacity, even though arching action is included in several expressions. The method that best predicted the shear capacity of SFRC is the expression provided by Kwak et al. [13] with an average tested/predicted value of 1.209 associated to a standard deviation of 0.421 and coefficient of variation of 34.8%. Nonetheless, Equation (12) provided by

Yakoub [34] had a good performance with an average tested/predicted value of 1.289, standard deviation of 0.214 and coefficient of variation of 16.6%; and the equation that provided the least variability is Equation (20) by Shin et al. [41], with a coefficient of variation of 15.2%, but it considerably overestimates the ultimate shear capacity of SFRC. For the inclined shear capacity, the best overall performance was provided by Equation (27) from Kwak et al. [13] with an average tested/predicted value of 1.661 associated to a coefficient of variation of 15.4%.

Finally, a fiber content of 1.2% or fiber factor 0.96 is calculated to replace the shear capacity provided by minimum stirrup according to ACI 318-14. For Eurocode 2, a 0.6% fiber content or fiber factor of 0.48 can be used to replace minimum stirrups as shear reinforcement. With these recommendations, the building industry can aim to use SFRC to replace minimum stirrups in regions with rebar congestion, and to take optimal advantage of the material properties. At the same time, further research on the different shear-carrying mechanisms and the underlying mechanics of the problem is necessary to theoretically support our experimental findings.

**Author Contributions:** Conceptualization, E.O.L.L.; methodology, E.O.L.L.; validation, E.O.L.L.; formal analysis, J.A.T.; investigation, J.A.T.; resources, E.O.L.L.; data curation, J.A.T.; writing—original draft preparation, J.A.T., E.O.L.L.; writing—review and editing, E.O.L.L.; visualization, J.A.T., E.O.L.L.; supervision, E.O.L.L.; project administration, E.O.L.L.; funding acquisition, E.O.L.L.

**Funding:** This research was funded by the program of Poligrants 2017–2018 and 2018–2019 from Universidad San Francisco de Quito. The APC was funded by the open access initiative of Delft University of Technology.

**Acknowledgments:** We would like to thank Bekaert for the donation of the fibers and Holcim for the donation of the aggregates. We appreciate the contributions of Santiago Hinojosa to the preparation of the experiments. For our work in the Materials Laboratory of Universidad San Francisco de Quito, we would like to thank Gustavo Tapia and Juan Jose Recalde for their invaluable support and suggestions during this research study.

**Conflicts of Interest:** The authors declare no conflict of interest.

## Notation

| | |
|---|---|
| $a/d$ | shear span to depth ratio |
| $b_w$ | width of the beam |
| $c$ | height of the compression zone |
| $d$ | effective depth |
| $d_a$ | maximum aggregate size |
| $d_v$ | internal lever arm |
| $f_c$ | concrete compressive strength |
| $f_c'$ | design concrete compressive strength |
| $f_{cuf}$ | cube compressive strength of steel fiber-reinforced concrete (SFRC) |
| $f_{ctf}$ | peak tensile stress of SFRC |
| $f_r$ | residual strength of SFRC |
| $f_{sp}$ | split tensile strength of SFRC |
| $f_y$ | longitudinal steel yield strength |
| $h$ | height of the cross section |
| $s_{xe}$ | equivalent crack spacing factor |
| $s_x$ | crack spacing parameter |
| $v_b$ | fiber contribution to shear strength |
| $v_{cr}$ | inclined shear capacity |
| $v_u$ | ultimate shear capacity |
| $w$ | crack width |
| $A_s$ | area of longitudinal steel reinforcement |
| $C_c$ | resultant of concrete under compression |
| $D$ | diameter of the fiber |
| $D_f$ | fiber bond factor = 1.00 for hooked fibers, 0.75 for crimped fibers, 0.5 for straight fibers |
| $E_{ct}$ | elastic modulus of SFRC in tension |
| $L$ | length of the fiber |
| $M$ | bending moment |

| | |
|---|---|
| $M_n$ | moment capacity of the cross section |
| $P$ | applied load |
| $P_{cr}$ | inclined cracking load |
| $P_u$ | ultimate load |
| $T_f$ | resultant of fibers under tension |
| $T_s$ | resultant of steel under tension |
| $V_c$ | shear force carried by the concrete |
| $V_{cr}$ | inclined cracking force |
| $V_f$ | fiber volume fraction |
| $V_{sf}$ | shear force carried by the steel fibers |
| $V_u$ | ultimate shear force |
| $\alpha$ | arching action factor for Khuntia et al. [45] |
| $\beta$ | factor that accounts for the strain at mid-depth and aggregate size for Yakoub |
| $\beta_1$ | Whitney's stress block coefficient |
| $\delta_u$ | deflection at ultimate load |
| $\varepsilon_{o85}$ | compressive strain measured at $0.85 f_c$ after peak |
| $\varepsilon_s$ | strain in longitudinal steel reinforcement |
| $\varepsilon_x$ | strain at mid-height of the cross section |
| $\varepsilon_{cu}$ | concrete ultimate strain |
| $\eta$ | factor that accounts for the effect of fiber in moment capacity |
| $\lambda$ | modification factor that accounts for the weight of the concrete |
| $\xi$ | size effect factor from Bažant and Kim [35] |
| $\rho$ | longitudinal reinforcement steel ratio |
| $\sigma_t$ | SFRC tensile stress |
| $(\sigma_t)_{avg}$ | SFRC average tensile stress |
| $\tau_{\max}$ | maximum bond strength of fiber-matrix interface |
| $\phi$ | strength reduction factor for ACI 318-14 [5] |
| $\psi$ | size effect factor from Imam et al. [33] |
| $\omega$ | reinforcement factor including fiber effect |
| $\theta$ | shear crack angle |

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
