# Peer review of "Influence of Fiber Content on Shear Capacity of Steel Fiber-Reinforced Concrete Beams"

_fibers, doi:10.3390/fib7120102_

Round 1

Reviewer 1 Report

The paper treats the influence of fibers content on shear capacity of steel fiber reinforced concrete beams. The paper is well written and can be accepted for publication after minor review.

Page 11: figure 6 is not mentioned in the text. Could the authors explain the strange initial slope of the load – displacement curves reported in figure 6?

Some editorial aspects:

Page 1 line 44: the sentence seems interrupted, after the word “when” something should be added.

Author Response

Reviewer 1

The paper treats the influence of fibers content on shear capacity of steel fiber reinforced concrete beams. The paper is well written and can be accepted for publication after minor review.

Thank you for your comments. We have provided an item-by-item response to your remarks here:

Page 11: figure 6 is not mentioned in the text. Could the authors explain the strange initial slope of the load – displacement curves reported in figure 6?

We had slip in the LVDT at the beginning of the load, after 8 kN ~ 10 kN a similar slope is observed. This is commented as follows:

“The resulting material properties from the concrete compressive and tensile strength tests are shown in Table 6 and a selection of load – displacement diagrams is shown in Figure 6, the difference in the slope of the first branch is due to slip of the LVDT at the beginning of the loading process”

Some editorial aspects:

Page 1 line 44: the sentence seems interrupted, after the word “when” something should be added.

Thanks for the observation, an explanation has been added in order to fix this sentence as follows:

“…but a minimum fiber content of 0.75% is permitted when the conditions provided in equation (1) are satisfied.”

Reviewer 2 Report

Please, refer to the attached pdf file

Round 2

Reviewer 2 Report

The Authors have addressed the Reviewer's comments in the revised manuscript.

In my opinion, the paper can be accepted in the present form.

Author Response

Thank you!